# The MIDAS domain of AAA mechanoenzyme Mdn1 forms catch bonds with two different substrates

Keith J Mickolajczyk[1], Paul Dominic B Olinares[2], Brian T Chait[2], Shixin Liu[3], Tarun M Kapoor[1]*

[1]Laboratory of Chemistry and Cell Biology, The Rockefeller University, New York, United States; [2]Laboratory of Mass Spectrometry and Gaseous Ion Chemistry, The Rockefeller University, New York, United States; [3]Laboratory of Nanoscale Biophysics and Biochemistry, The Rockefeller University, New York, United States

**Abstract** Catch bonds are a form of mechanoregulation wherein protein-ligand interactions are strengthened by the application of dissociative tension. Currently, the best-characterized examples of catch bonds are between single protein-ligand pairs. The essential AAA (ATPase associated with diverse cellular activities) mechanoenzyme Mdn1 drives at least two separate steps in ribosome biogenesis, using its MIDAS domain to extract the ubiquitin-like (UBL) domain-containing proteins Rsa4 and Ytm1 from ribosomal precursors. However, it must subsequently release these assembly factors to reinitiate the enzymatic cycle. The mechanism underlying the switching of the MIDAS-UBL interaction between strongly and weakly bound states is unknown. Here, we use optical tweezers to investigate the force dependence of MIDAS-UBL binding. Parallel experiments with Rsa4 and Ytm1 show that forces up to ~4 pN, matching the magnitude of force produced by AAA proteins similar to Mdn1, enhance the MIDAS domain binding lifetime up to 10-fold, and higher forces accelerate dissociation. Together, our studies indicate that Mdn1's MIDAS domain can form catch bonds with more than one UBL substrate, and provide insights into how mechanoregulation may contribute to the Mdn1 enzymatic cycle during ribosome biogenesis.

*For correspondence:
kapoor@mail.rockefeller.edu

**Competing interest:** The authors declare that no competing interests exist.

## Editor's evaluation

Mickolajczyk et al., use solution and single-molecule approaches to characterize the binding of the MIDAS domain from the ribosome maturation factor Mdn1 with ubiquitin-like domains from two assembly factors, Ytm1 and Rsa4. Both interactions are specific but weak in solution. A clever experimental setup allows the authors to also measure the interaction with optical tweezers, revealing catch and slip bond modes, depending on the applied load. The off-rate is lowest at ~4 pN. The behavior might help to explain how Mdn1 binding to Ytm1 and Rsa4 (and possibly other UBL-containing proteins) is stable enough for protein extraction from pre-60S particles without excessive idle binding of free assembly factors.

## Introduction

Regulation of protein-ligand interactions is an essential organizational principle in cell biology, and is achieved through various means, including post-translational modifications (e.g. phosphorylation) and substrate exchange (e.g. GTP hydrolysis) (*Alberts et al., 1994*). More recently, mechanoregulation – or changes in protein-ligand interactions in response to applied mechanical forces – has garnered interest as another organizing principle. While most protein-ligand interactions are known to become

**Figure 1.** Solution measurements of MIDAS-Rsa4 interaction. (**A**) Domain diagrams (drawn to scale) of full-length Mdn1 (top), and the SNAP-tagged MIDAS constructs (bottom). (**B**) Domain diagrams (drawn to scale) highlighting the position of the ubiquitin-like (UBL) and WD repeat domains within the assembly factor Rsa4, as well as the added green fluorescent protein (GFP) tag. (**C**) SDS-PAGE gel (Coomassie staining) of final purified Rsa4-GFP. (**D**) Native mass spectrometry analysis of Rsa4-GFP. Expected mass (with loss of N-terminal methionine) 83.962 kDa (red dotted line), measured mass 83.981 ± 0.001 kDa (mean ± standard deviation, SD). Full spectrum in *Figure 1—figure supplement 1*. (**E**) Elution profile (monitored by GFP fluorescence) of Rsa4-GFP (40 µM prior to injection) either alone or pre-mixed with MIDAS-WT or MIDAS-Y4666R (no GFP label; 60 µM) on a Superdex 200 Increase size exclusion column. (**F**) SDS-PAGE gels (Coomassie staining) corresponding to the elution profiles shown in panel E. See also *Figure 1—figure supplement 2*. (**G**) Microscale thermophoresis data showing the binding of MIDAS protein to Rsa4-GFP (50 nM). All data shown as mean ± SD for n = 4 independent experiments including at least two separate preparations of each protein used. The MIDAS-WT data were fitted to a binding isotherm (black curve). MIDAS-Y4666R data connected by lines to guide the eye.

The online version of this article includes the following source data and figure supplement(s) for figure 1:

**Source data 1.** Source data for size exclusion chromatography, microscale thermophoresis (MST), and mass photometry assays shown in *Figure 1* and its supplements.

**Source data 2.** All raw TIFF files for SDS-PAGE gels corresponding to size exclusion chromatography coelution assays.

**Figure supplement 1.** Preparation of recombinant Rsa4-GFP.

**Figure supplement 2.** Full gels for Rsa4-GFP size exclusion chromatography experiments.

weaker when external forces are applied along the bond axis, referred to as slip bonds, a few special cases have been identified where force increases bond lifetimes, referred to as catch bonds (*Bell, 1978*; *Sokurenko et al., 2008*). Catch bonds are hypothesized to be a widely used regulatory mechanism in the cell (*Sokurenko et al., 2008*), but force spectroscopy investigations to date have largely focused on cell adhesion and cytoskeleton proteins. Moreover, most known examples of catch bonds are for one protein interacting with one specific substrate.

Ribosome biogenesis is a complex, multi-step process involving hundreds of trans-acting protein and RNA factors, including ATP-driven mechanoenzymes (*Frazier et al., 2021*; *Kressler et al., 2010*). One essential mechanoenzyme is Mdn1, a member of the AAA (ATPase associated with diverse cellular activities) superfamily. Mdn1 is a large (>500 kDa) multi-domain enzyme; at its N-terminus are six AAA domains, which fold into a pseudohexameric ring, followed by a linker including both a structured (~1800 amino acids) and a likely non-structured (~517 amino acids) region, and finally a C-terminal (290 amino acid) MIDAS domain (*Figure 1A*; *Garbarino and Gibbons, 2002*). Mdn1 binds and extracts the ubiquitin-like (UBL) domain-containing assembly factors Ytm1 and Rsa4, which are embedded in pre-60S (large) ribosomal precursors at different stages of maturation, via its MIDAS domain (*Bassler et al., 2010*; *Ulbrich et al., 2009*). Recent structural work revealed that the MIDAS

domain docks onto the AAA ring (*Chen et al., 2018*; *Sosnowski et al., 2018*) in the context of pre-60S binding, and forms a tripartite connection in which it is also bound to Rsa4 ( *Kater et al., 2020*). However, we do not understand how the MIDAS-UBL binding, which must be strong for assembly factor removal, switches to a more weakly bound state, such that the UBL proteins can be subsequently released.

The Mdn1 MIDAS domain bears structural homology to the integrin α I domain (*Garbarino and Gibbons, 2002*). In solution, this integrin domain has a weak (~1 mM) affinity for its ligands (*Shimaoka et al., 2001*). Remarkably, this affinity can be enhanced by about an order of magnitude by applied force (*Astrof et al., 2006*; *Kong et al., 2009*; *Shimaoka et al., 2003*). Although a UBL domain is structurally distant from extracellular integrin ligands, it is possible that Mdn1 may similarly use mechanoregulation for its function. Consistent with this idea, Brownian dynamics simulations have shown that docking of the MIDAS domain onto the AAA ring stretches the unstructured portion of the linker such that 1–2 pN of tension is generated (*Mickolajczyk et al., 2020*). This tension would be transmitted along the MIDAS-UBL bond axis, and its magnitude may even be enhanced by ATP-dependent motions propagated from the AAA ring (*Chen et al., 2018*; *Ulbrich et al., 2009*). However, no studies to date have tested the hypothesis that the Mdn1 MIDAS domain can form catch bonds with its substrates.

In the current work, we investigate the mechanoregulation of the interaction between Mdn1's MIDAS domain and two UBL domain-containing assembly factors. Using bulk assays, we find that MIDAS-Rsa4 and -Ytm1 affinity is weak (≥7 µM). Using an optical tweezers 'force jump' assay, we find that the Mdn1 MIDAS domain forms a catch bond with both Rsa4 and Ytm1, with bond lifetimes extending ~10-fold at ~4 pN tension. These measurements show that mechanical forces are sufficient for modulating Mdn1 substrate binding, and suggest that catch bonds play a key role in regulating Mdn1-driven steps in ribosome biogenesis.

## Results

### Solution measurements of MIDAS-Rsa4 binding

To investigate the regulation of Mdn1 binding to its UBL domain-containing pre-ribosomal substrate Rsa4, we first sought to characterize binding in vitro. The Mdn1 MIDAS domain (amino acids 4381–4717) with an N-terminal SNAP tag was expressed in *Escherichia coli* as before (*Figure 1A*; *Mickolajczyk et al., 2020*). Both wild-type MIDAS domain (MIDAS-WT) and MIDAS domain harboring a point mutation known to diminish UBL domain binding (MIDAS-Y4666R) were generated (*Ahmed et al., 2019*). Full-length Rsa4 with a C-terminal green fluorescent protein (GFP) (Rsa4-GFP; *Figure 1B*) was expressed in insect cells and purified using affinity, ion exchange, and size exclusion chromatography (see Materials and methods). Purity was assessed by SDS-PAGE (*Figure 1C*, *Figure 1—figure supplement 1A*). Rsa4-GFP was confirmed to be a monomer of the correct molecular weight both by native mass spectrometry (nMS) (to within 20 Da; *Figure 1D*, *Figure 1—figure supplement 1B*) and by mass photometry (to within 2 kDa; *Figure 1—figure supplement 1C*).

Binding of the Mdn1 MIDAS domain to Rsa4-GFP was first assessed by size exclusion chromatography coelution assays. Rsa4-GFP (40 µM prior to injection) by itself eluted as a single peak at ~14.5 mL, as measured by GFP fluorescence (*Figure 1E*) and SDS-PAGE (*Figure 1F*, black; uncropped gels in *Figure 1—figure supplement 2*). MIDAS-WT alone and MIDAS-Y4666R alone (60 µM prior to injection) did not produce a fluorescence signal, but were each seen to elute at ~14.3 mL by SDS-PAGE (*Figure 1F*, gray). Mixing Rsa4 with MIDAS-WT before injection led to a shift in the fluorescence profile toward a higher molecular weight, with an overlapping elution profile consistent with binding (*Figure 1E–F*, blue). Mixing Rsa4 with MIDAS-Y6664R before injection did not lead to a shift in the fluorescence profile or to apparent coelution by SDS-PAGE (*Figure 1E–F*, red), suggesting that the coelution with MIDAS-WT is due to specific binding.

We next sought to determine the affinity of the Mdn1 MIDAS domain for Rsa4-GFP using microscale thermophoresis (MST). Titrating MIDAS-WT against Rsa4-GFP (50 nM) produced changes in the normalized fluorescence signal upon heating ($|\Delta F_{Norm}|$) that could be fitted to a binding isotherm with $K_D = 6.9 \pm 2.0$ µM (fit ±95% confidence intervals, CI) (*Figure 1G*). Titrating MIDAS-Y4666R led to smaller $|\Delta F_{Norm}|$ values that could not be fitted, consistent with weaker binding. Together, these results

show that while the Mdn1 MIDAS domain can bind to Rsa4 in solution, the binding affinity is weak (~7 μM).

## Single-molecule measurements of MIDAS-Rsa4 binding under applied load

To investigate the force dependence of the MIDAS-UBL domain interaction, we developed a single-molecule optical tweezers 'force jump' assay (*Figure 2A*). In this assay, two double-stranded (dsDNA) handles are attached to two polystyrene beads – one held in a micropipette by suction and the other held by an optical trap – and connected using a single-stranded (ssDNA) bridge. SNAP-tagged MIDAS and GFP nanobody proteins are covalently bound to ssDNA oligonucleotides (*Figure 2B*), which are then annealed to the bridge strand (*Figure 2C*). Rsa4-GFP in solution can bind to the GFP nanobody on the bridge construct. With the tethered assembly in place, the optical trap, in constant force mode, can be rapidly switched (i.e. force jumped) between a low and a high constant force. At the high force, MIDAS and Rsa4-GFP are physically separated and cannot bind (*Figure 2A* state 1). At the low force (*Figure 2A* state 2) the bridge strand is relaxed and MIDAS and Rsa4 come into close enough proximity to bind. Should the proteins bind while at the low force, an intermediate position can be read out when the system is jumped to high force (*Figure 2A* state 3). This intermediate position informs on the lifetime of the MIDAS-Rsa4 interaction. The distance traveled between the intermediate and final high-force position (Δx) should depend on the length of the single-stranded region of annealed bridge construct. We thus performed our assays with two bridge constructs of different lengths (55 and 70 nt; *Figure 2C*). For both bridge constructs, we ensured that only a single 'tether' was drawn between each pair of beads by examining the shape of the force-extension curve and matching it to a known hairpin standard (*Figure 2D*, see Materials and methods).

Example data generated in the force jump assay are shown in *Figure 2E–G*. Here, we held a constant force of 0.5 pN for at least 5 s before jumping to a constant force of 6 pN, also held for at least 5 s. When no GFP-tagged assembly factor was present (negative control), only two positional states were detected (here using the 55 nt bridge construct; *Figure 2E*). When Rsa4-GFP was added (20 nM) an intermediate position appeared on some of the jumps, consistent with MIDAS-Rsa4 binding (*Figure 2F*). For these intermediate positions, we quantified both a bond lifetime ($\tau$) and a distance change (Δx) using a two-state hidden Markov model algorithm (see Materials and methods). We next made measurements using the 70 nt bridge construct, and again saw intermediate positions, but with larger Δx magnitudes (*Figure 2G*). For both the 55 and 70 nt bridge constructs, intermediate positions were not seen in the negative control (*Figure 2—figure supplement 1*). Altogether, these data exemplify single-molecule measurements of MIDAS-Rsa4 binding under applied forces.

## Force dependence of MIDAS-Rsa4 binding

We next sought to measure the force dependence of binding by quantifying bond lifetimes ($\tau$) of MIDAS and Rsa4-GFP on both bridge constructs (55 and 70 nt) at multiple high-force levels (hereafter referred to as total applied force levels, $F_{Tot}$). We first measured MIDAS-WT with Rsa4 on the 55 nt bridge construct at 6 pN (*Figure 3A*). The distribution (n = 57 events from 14 molecules) of bond lifetimes could be fitted to a single exponential (appears linear on a semilog plot) as opposed to a higher-order exponential (*Figure 3—figure supplement 1*), consistent with the kinetics of exit from a single bound state (*Guo and Guilford, 2006*). The inverse of the average bond lifetime is equivalent to the off-rate. As an additional control, we measured the bond lifetime of Rsa4-GFP with oligonucleotide-bound MIDAS-Y4666R, the weak-binding mutant (also run on the 55 nt tether construct with an $F_{Tot}$ value of 6 pN). This data (n = 29 events from 17 molecules) could also be described with a single exponential. The substantially shorter bond lifetime distribution of MIDAS-Y4666R with Rsa4-GFP versus MIDAS-WT provides evidence that the binding events measured are specific.

We next built distributions for the bond lifetime of Mdn1 MIDAS-WT with Rsa4-GFP at multiple values of $F_{Tot}$ on both the 55 (*Figure 3B*) and 70 nt (*Figure 3C*) tether constructs (all distributions in *Figure 3—figure supplements 2–3*). In each of these experiments, we observed that the distribution shifted from short to long bond lifetimes between 4 and 6 pN, and then back to short bond lifetimes at 12 pN. All measured distributions could be fitted to a single exponential (residuals and analysis in *Figure 3—figure supplements 2–3*), indicating that applied force influenced the kinetics of dissociation, not the number of states from which dissociation could occur (*Huang et al., 2017*). Plotting

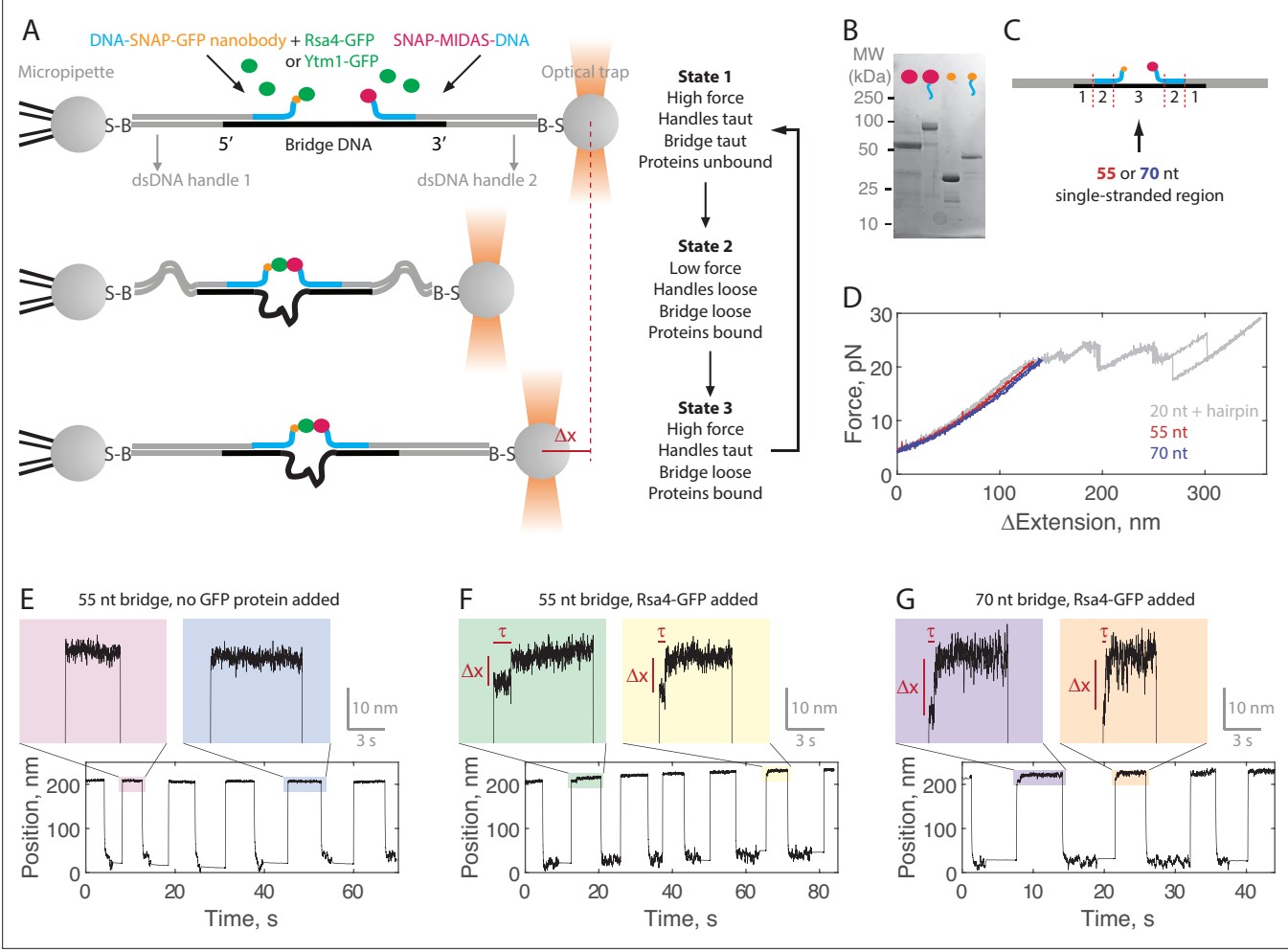

**Figure 2.** Single-molecule assay for measuring the MIDAS-Rsa4 interaction under load. (**A**) Single-molecule optical tweezers 'force jump' assay design. Two 1.5 kilobase-pair double-stranded (dsDNA) handles are attached to 2.1 μm beads via biotin-streptavidin (**B–S**) linkage. The handles have 31–33 nucleotide (nt) overhangs, to which a single-stranded (ssDNA) 'bridge' is annealed. A copy of green fluorescent protein (GFP) nanobody and MIDAS protein, each conjugated to a DNA oligonucleotide (blue), are then annealed to the bridge. Rsa4-GFP, free in solution, can bind to the GFP nanobody. Dissociation of the MIDAS-UBL interaction under load can be read out as a positional change (Δx) when the optical trap (in constant force mode) is switched from a low to a high applied force. (**B**) SDS-PAGE gel (Coomassie staining) showing the covalent attachment of ssDNA oligonucleotides to the SNAP-tagged MIDAS-WT (pink) and SNAP-tagged GFP nanobody (orange). (**C**) Sections of the DNA bridge. Section 1 (31–33 nt) anneals to the dsDNA handles, section 2 (30 nt) anneals the protein-bound DNA oligonucleotides, and section 3 remains single-stranded. Different lengths of section 3 (55 or 70 nt) are expected to produce Δx events of different magnitude. (**D**) Example force-extension curves of the DNA handles connected by the 55 nt (red) and 70 nt (blue) bridge. Also shown is a hairpin (gray) which anneals to the dsDNA handles with a 20 nt region leftover; here the distinct unfolding/refolding pattern at loads above 20 pN enables identification of a single 'tether' (two dsDNA handles connected by a bridge) between the two beads. Data generated on tethers whose force-extension curves did not overlap with that of the hairpin standard in the 4–20 pN (ΔExtension = 0 when Force = 4 pN) range were not used. (**E**) Example force jump data (0.5 and 6 pN low and high force) on the 55 nt bridge with all components added except a GFP-labeled assembly factor. In some instances, force feedback was released at the low force level to reduce large fluctuations. Only two position levels were observed. (**F**) Example force jump data on the 55 nt bridge with Rsa4-GFP (20 nM) added. On some jumps (highlighted), an intermediate position is observed. (**G**) Similar to panel F, with the 70 nt bridge construct.

The online version of this article includes the following source data and figure supplement(s) for figure 2:

**Source data 1.** Source data for example force jump plots.

**Figure supplement 1.** Fraction of active tethers.

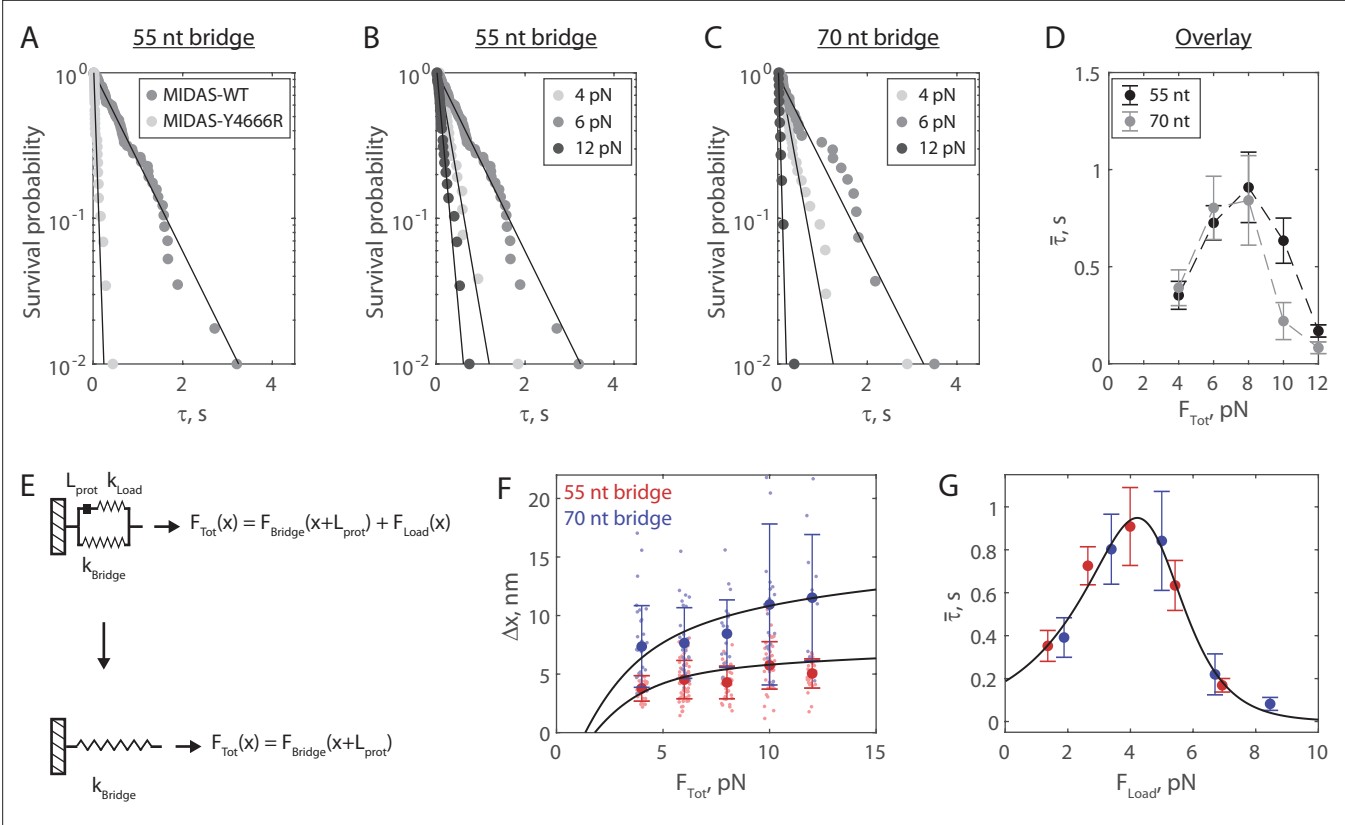

**Figure 3.** The Mdn1 MIDAS domain forms a catch bond with Rsa4. (**A**) Distributions of MIDAS-Rsa4 bond lifetimes (6 pN total force, 55 nt bridge construct) with either MIDAS-WT (n = 57 events from 14 molecules) or MIDAS-Y4666R (n = 29 events from 17 molecules). Survival probability is defined as one minus the empirical cumulative density function. For all distributions, the final data point was moved from y = 0 to y = 0.01 to enable semilog plotting. Black lines show fits to a single exponential. (**B**) Distributions of MIDAS-Rsa4 bond lifetimes on the 55 nt bridge construct at 4 pN (n = 26 events from eight molecules), 6 pN (n = 57 events from 14 molecules), and 12 pN (n = 29 events from five molecules) total applied force. (**C**) Distributions of MIDAS-Rsa4 bond lifetimes on the 70 nt bridge construct at 4 pN (n = 33 events from 11 molecules), 6 pN (n = 27 events from seven molecules), and 12 pN (n = 11 events from five molecules) total applied force. (**D**) The average bond lifetime of MIDAS-Rsa4 binding as a function of total applied force. Data shown as mean ± standard error of the mean (SEM; n = 26–57 events from 38 total molecules for the 55 nt bridge and n = 11–33 events from 28 total molecules; note that events at more than one force could be collected on a given molecule), with dotted lines to guide the eye. (**E**) Mechanical circuit model describing the force jump assay. When the proteins (of inextensible length $L_{prot}$) are bound, force is partitioned between the top 'loading' strand (two 12 nt single-stranded regions) and the bottom bridge strand. In the equations shown inset, x designates extension along the loading stand spring. (**F**) The magnitude of Δx for Rsa4-MIDAS interactions as a function of total applied force. Individual measurements shown as small data points, mean ± SD (n = 11–57 events) shown in bold. Data generated with the 55 and 70 nt bridge constructs shown in red and blue, respectively. Black lines show output of the mechanical circuit model (see also **Figure 3—figure supplement 4**). (**G**) The average MIDAS-Rsa4 bond lifetime as a function of force applied across the proteins. Data generated with the 55 nt bridge construct shown in red and data generated with the 70 nt bridge construct shown in blue. Data points shown as mean ± SEM (n = 11–57 events). Black curve shows fit to the catch-slip Bell model.

The online version of this article includes the following source data and figure supplement(s) for figure 3:

**Source data 1.** Source data for data generated in the force jump assay shown in **Figure 3** and its supplements.

**Source data 2.** MATLAB script that runs the mechanical circuit model of the force jump assay.

**Source data 3.** Raw TIFF file for SDS-PAGE gel of Rsa4-SNAP.

**Figure supplement 1.** Comparing single and double exponential fits for bond lifetime data.

**Figure supplement 2.** Complete force jump dataset for Rsa4-GFP on 55 nt bridge construct.

**Figure supplement 3.** Complete force jump dataset for Rsa4-GFP on 70 nt bridge construct.

**Figure supplement 4.** Mechanical circuit model for the force jump assay.

**Figure supplement 5.** Force jump assay using Rsa4-SNAP.

**Figure supplement 6.** Complete force jump dataset for Rsa4-SNAP on 55 nt bridge construct.

the average bond lifetime ($\bar{\tau}$) as a function of $F_{Tot}$, we noted a chevron-shaped curve for both bridge constructs with minor differences at low and high $F_{Tot}$ values (*Figure 3D*). The average bond lifetime increased between $F_{Tot}$ values of 4–8 pN, and then decreased at $F_{Tot}$ values of 10–12 pN. This stabilization of binding by external load up to a certain threshold (catch bond) contrasts the more common scenario where external force monotonically accelerates dissociation (slip bond).

## Mdn1 MIDAS and Rsa4 form a catch bond

In our experimental geometry, the total force is partitioned between the protein pair and the bridge DNA (*Figure 2A*). To derive the force applied to the protein bond from $F_{Tot}$, we built a mechanical circuit model of the force jump assay (*Figure 3E*, *Figure 3—figure supplement 4*). When MIDAS and Rsa4-GFP are bound, we model the system as two nonlinear springs in parallel. The top (loading) spring consists of the 12 nt sections of the oligonucleotides bound to SNAP-GFP nanobody and SNAP-MIDAS that do not anneal to the bridge DNA, as well as the relatively rigid folded proteins, which we model as inextensible (constant length $L_{prot}$) at 8 nm (*Ahmed et al., 2019*). The bottom spring is the non-complementary ssDNA region of the bridge DNA (55 or 70 nt). Since the proteins are on the top spring, the bottom spring begins pre-stretched relative to the top spring by $L_{prot}$. In this configuration, the total force applied by the optical trap ($F_{Tot}$) is partitioned over the two springs ($F_{Load}$ and $F_{Bridge}$). When MIDAS and Rsa4-GFP dissociate from one another, the entirety of $F_{Tot}$ is put onto the bridge DNA, leading to further stretching ($\Delta x$). We modeled the nonlinear springs in the system using the worm-like chain equation, a model for the elasticity of DNA (see Materials and methods). An overlay of experimental measurements of $\Delta x$ with output of the mechanical circuit model is shown in *Figure 3F*. We note that the model output (black lines) is not fitted to or constrained by the experimental data. The agreement between theory and experiment provides further validation of the force jump assay and the mechanical circuit model.

We next used the mechanical circuit model to combine the bond lifetime data generated using the 55 and 70 nt bridge constructs (*Figure 3D*) and plot them as a function of $F_{Load}$, the force actually placed on the protein-protein interaction (*Figure 3G*). The combined data smoothed the chevron shape, and could be fitted to the 'catch-slip' application of the Bell model (*Barsegov and Thirumalai, 2005*; *Evans et al., 2004*; *Evans and Ritchie, 1997*; *Guo and Guilford, 2006*):

$$\bar{\tau}^{-1} = k_c^0 exp\left(\frac{F \cdot x_c}{k_B T}\right) + k_s^0 exp\left(\frac{F \cdot x_s}{k_B T}\right)$$

where $\bar{\tau}$ is the average bond lifetime and subscripts c and s refer to the catch and slip pathways, respectively. Here, the catch pathway dominates at low forces and the slip pathway dominates at high forces, respectively generating the rise and fall of the chevron shape. We estimated parameters $k_c^0 = 5.3 \pm 2.6$ s$^{-1}$, $x_c = -2.0 \pm 0.8$ nm, $k_s^0 = 0.01 \pm 0.02$ s$^{-1}$, and $x_s = 4.1 \pm 1.6$ nm (fit ±95% CI) for MIDAS with Rsa4-GFP. The similar values of the distance parameters $x_c$ and $x_s$ give rise to a roughly symmetric rise and fall in force-dependent bond lifetime, similar to the case of the P-selectin complex with sPSGL-1 (*Barsegov and Thirumalai, 2005*). Additionally, this fit allowed us to determine the critical force at which bond lifetime is maximized, which we estimated to be 4.2 pN. As an additional control, we repeated the force jump experiments using Rsa4 with a C-terminal SNAP tag (Rsa4-SNAP), which was directly conjugated to an ssDNA oligonucleotide. Datasets generated with Rsa4-GFP and Rsa4-SNAP were in good agreement with each other (*Figure 3—figure supplements 5–6*), indicating that the GFP-GFP nanobody connection did not influence bond lifetime measurements. Overall, this analysis shows that the Mdn1 MIDAS domain forms a catch bond with Rsa4.

## Mdn1 MIDAS also forms a catch bond with Ytm1

We next tested whether Mdn1 MIDAS catch bond behavior is specific for Rsa4, or more general with another UBL domain-containing ribosome assembly factor. Mdn1 has been proposed to also remove Ytm1, from the pre-60S particle (*Ahmed et al., 2019*; *Bassler et al., 2010*). However, the *Schizosaccharomyces pombe* Rsa4 and Ytm1 UBL domains only have 43.0% sequence similarity. Ytm1-GFP (*Figure 4A*) was generated in insect cells similarly to Rsa4-GFP. Ytm1-GFP ran with an apparent double-banding pattern (*Figure 4A*), however the protein was assessed to be homogenous, monomeric, and centered around the expected molecular weight by both nMS (to within 20 Da) and mass photometry (to within 1 kDa; *Figure 4—figure supplement 1*). We ran size exclusion chromatography assays, and as with Rsa4-GFP, coelution was seen for Ytm1-GFP with MIDAS-WT but not

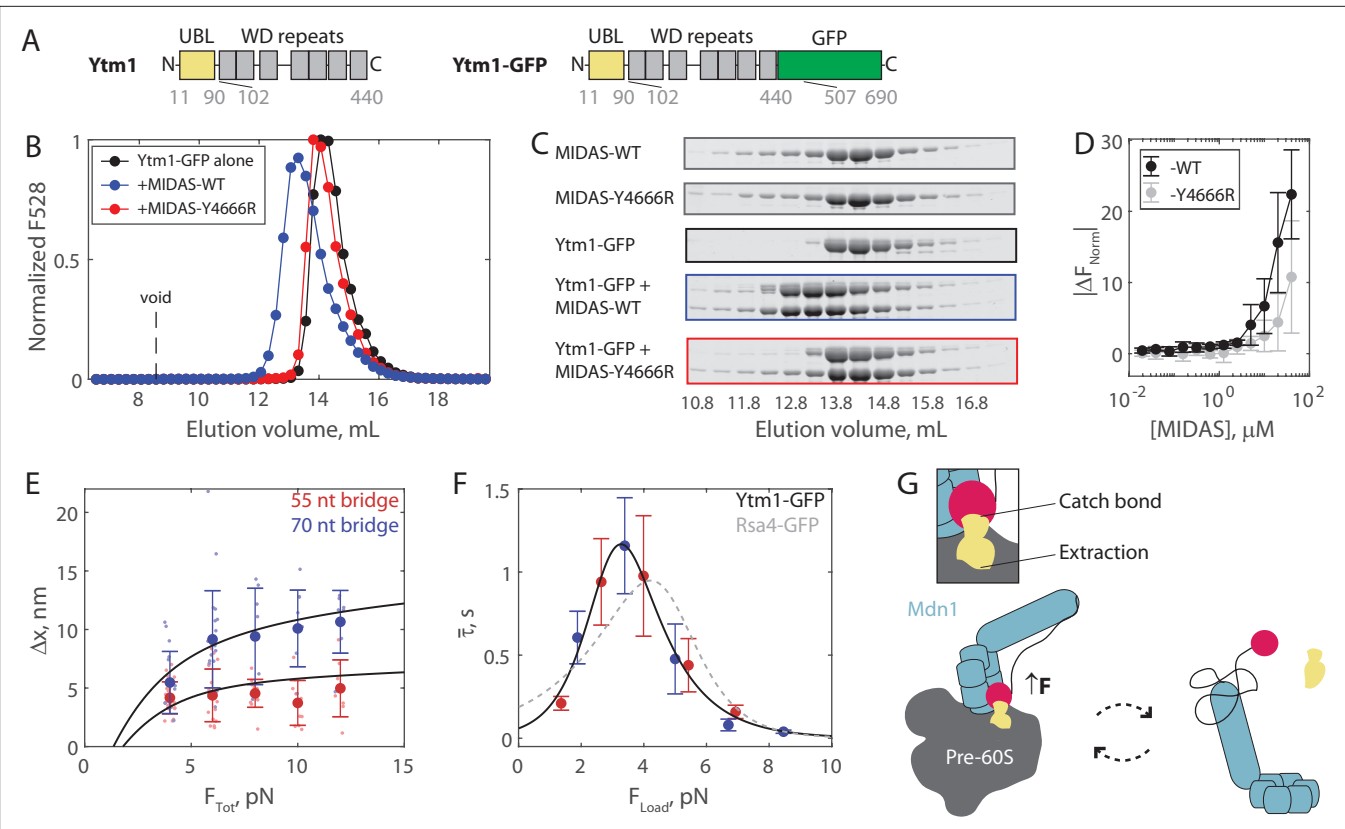

**Figure 4.** The Mdn1 MIDAS domain also forms a catch bond with Ytm1. (**A**) Domain diagrams (drawn to scale) highlighting the position of the ubiquitin-like (UBL) and WD repeat domains within the assembly factor Ytm1, as well as the added green fluorescent protein (GFP) tag. (**B**) Elution profile (monitored by GFP fluorescence) of Ytm1-GFP (40 µM prior to injection) either alone or pre-mixed with MIDAS-WT or MIDAS-Y4666R (no GFP label; 60 µM) on a Superdex 200 Increase size exclusion column. (**C**) SDS-PAGE gels (Coomassie staining) corresponding to the elution profiles shown in panel B. (**D**) Microscale thermophoresis data showing the binding of unlabeled MIDAS protein to Ytm1-GFP (50 nM). All data shown as mean ± SD (n = 4). Data points connected by lines to guide the eye. (**E**) The magnitude of Δx for Ytm1-MIDAS interactions as a function of total applied force. Individual measurements shown as small data points, mean ± SD (n = 8–22 events from 30 total molecules on the 55 nt tether construct, and n = 8–22 events from 24 total molecules on the 70 nt tether construct) shown in bold. Black lines show output of the mechanical circuit model. Example traces with Ytm1-GFP in the force jump assay shown in ***Figure 4—figure supplement 3***. (**F**) The average bond lifetime of MIDAS-Ytm1 binding as a function of force applied across the proteins. Data generated with the 55 nt bridge construct shown in red and data generated with the 70 nt bridge construct shown in blue. Data points shown as mean ± SEM (n = 8–22 events from 30 total molecules on the 55 nt tether construct, and n = 8–22 events from 24 total molecules on the 70 nt tether construct). All distributions in ***Figure 4—figure supplements 4 and 5***. Black curve shows fit to the catch-slip Bell model. Gray dotted line shows the fit for MIDAS-Rsa4 binding (***Figure 3G***) for comparison. (**G**) Two-state diagram highlighting the MIDAS-UBL catch bond in ribosome biogenesis. Force generated by Mdn1 (blue) must be transmitted through its MIDAS domain (magenta) to mechanochemically remove an assembly factor, Rsa4 or Ytm1 (yellow), from the pre-60S particle (gray). One possibility is that stretching of the unstructured Mdn1 linker (black) builds tension (black arrow), which is transmitted along the MIDAS-UBL bond axis when the assembly factor is embedded in the pre-60S, engaging the catch bond (left side). Inset: tension strengthens the MIDAS-assembly factor bond, while simultaneously promoting assembly factor extraction from the pre-60S. Tension is relieved upon release from the pre-60S, disengaging the catch bond and thus promoting MIDAS-UBL dissociation (right side).

The online version of this article includes the following source data and figure supplement(s) for figure 4:

**Source data 1.** Source data for size exclusion chromatography, microscale thermophoresis (MST), and force jump assays shown in ***Figure 4*** and its supplements.

**Source data 2.** All raw TIFF files for SDS-PAGE gels corresponding to size exclusion chromatography coelution assays.

**Figure supplement 1.** Preparation of recombinant Ytm1-GFP.

**Figure supplement 2.** Full gels for Ytm1-GFP size exclusion chromatography experiments.

**Figure supplement 3.** Force jump assay with Ytm1-GFP.

**Figure supplement 4.** Complete force jump dataset for Ytm1-GFP on 55 nt bridge construct.

**Figure supplement 5.** Complete force jump dataset for Ytm1-GFP on 70 nt bridge construct.

with MIDAS-Y4666R (*Figure 4B–C*, *Figure 4—figure supplement 2*). We also used MST to assess MIDAS-Ytm1-GFP binding. Titrating MIDAS-WT against Ytm1-GFP (50 nM) led to larger |ΔF$_{Norm}$| values than titrating MIDAS-Y4666R, but the data could not be fitted to a binding isotherm in either case (*Figure 4D*). Hence. MIDAS and Ytm1-GFP can bind in solution, but interaction is even weaker than that of MIDAS and Rsa4-GFP.

We next ran the force jump assay with Ytm1-GFP present instead of Rsa4-GFP. We measured bond lifetimes and Δx values using both the 55 and 70 nt bridge constructs, and measured a force dependence of binding (*Figure 4—figure supplements 3–5*). Measurements of Δx made with Ytm1-GFP present fell along the expected curves based on the mechanical circuit model (without any free parameter), similar to Rsa4-GFP (*Figure 4E*). Plotting the average bond lifetime as a function of F$_{Load}$, the force experienced by the proteins in the force jump assay, we found that Mdn1 MIDAS can form a catch bond with Ytm1-GFP (*Figure 4F*). The fitted parameters were similar to those of Rsa4-GFP: $k_c^0$ = 15.2 ± 20.9 s$^{-1}$, $x_c$ = –4.8 ± 2.8 nm, $k_s^0$ = 0.04 ± 0.05 s$^{-1}$, and $x_s$ = 3.2 ± 0.9 nm (fit ±95% CI). The critical force where binding lifetime is maximized was 3.3 pN, versus 4.2 pN for Rsa4-GFP. Overall, these results show that the Mdn1 MIDAS domain forms a catch bond with two different UBL domain-containing proteins.

## Discussion

In the current work, we investigate the regulation of binding between the Mdn1 MIDAS domain and the UBL domain-containing ribosome assembly factors Rsa4 and Ytm1. We show that the Mdn1 MIDAS domain has weak affinity for Rsa4 and Ytm1 in solution (≥7 μM). Using a single-molecule optical tweezers force jump assay developed for this study, we show that Mdn1 MIDAS forms a catch bond with both Rsa4 and Ytm1, with bond lifetimes increasing by an order of magnitude under applied tension up to ~4 pN, and decreasing back to baseline between ~4 and 10 pN. Together, our findings provide insights into how forces may directly regulate Mdn1-driven steps in the nucleolus and nucleoplasm during ribosome biogenesis.

Based on our findings, we propose a model where the Mdn1 MIDAS-UBL catch bond plays a key role in assembly factor processing (*Figure 4G*). Mdn1 binds the pre-60S particle and establishes a tripartite interaction between its AAA ring, its MIDAS domain, and the UBL domain of the assembly factor substrate that it must extract (*Chen et al., 2018*; *Kater et al., 2020*; *Sosnowski et al., 2018*). In this MIDAS-docked state, Mdn1's unstructured linker, which acts as an entropic spring, is stretched; computational modeling of MIDAS docking estimates that 1–2 pN of tension is built up and propagated along the MIDAS-UBL bond axis (*Mickolajczyk et al., 2020*). A catch bond mechanism helps to explain how this tension promotes, rather than disfavors, MIDAS-UBL binding; 1–2 pN is below the ~3.3–4.2 pN peak of the MIDAS-UBL catch bond, but one possibility is that additional tension is generated by Mdn1's ATPase activity. The MIDAS domain remains bound to the UBL domain until the assembly factor dissociates from the pre-60S particle. Subsequently, the MIDAS domain undocks from the AAA ring, relieving tension across the bond with the UBL domain, deactivating the catch bond and facilitating assembly factor release. Ytm1 and Rsa4 removal occur at separate stages of pre-60S maturation, in the nucleolus and nucleoplasm, respectively (*Bassler et al., 2010*; *Kressler et al., 2012*). We propose that our model holds for both these scenarios. We also note that free copies of Ytm1 and Rsa4 are present in the nucleolus and nucleoplasm (*Kressler et al., 2012*); the Mdn1 catch bond thus also prevents Mdn1-Rsa4/Ytm1 binding while off the pre-60S, which would compete away usable copies of each protein. Currently, the kinetics of Mdn1-driven steps in ribosome assembly are not well established. Future work is thus necessary to place the ~1 s bond lifetimes measured here within the multi-step enzymatic process of ribosome biogenesis.

It is interesting to note that Mdn1 processes UBL domain-containing proteins. The UBL domains of Rsa4 and Ytm1 have low sequence similarity (43.0% in *S. pombe*), but a high level of structural similarity (*Ahmed et al., 2019*; *Romes et al., 2016*). There are numerous (>10 in *S. pombe*, >60 in humans) other UBL domain-containing proteins with similar folds to Ytm1/Rsa4-UBL (*Collins and Goldberg, 2020*; *Hartmann-Petersen and Gordon, 2004*). Moreover, UBL domains can be covalently conjugated to other proteins by a cascade of UBL-specific enzymes (*Schulman, 2011*; *Streich and Lima, 2014*). In addition, Mdn1 in mammalian cells has been shown to be targeted to SUMOylated substrates (*Raman et al., 2016*), where SUMO and UBL domains also share structural similarity. While no Mdn1 MIDAS-binding partners other than Rsa4 and Ytm1 have been confirmed to date, it is

noteworthy that Mdn1 was recently implicated to also have a role in heterochromatic RNA clearance (*Shipkovenska et al., 2020*). Therefore, we speculate that Mdn1 may also process UBL or SUMO domain-containing proteins other than Rsa4 and Ytm1 in the cell. Based on the close similarity of the Rsa4 and Ytm1 catch bonds measured, we predict that the Mdn1 MIDAS domain would also form catch bonds with these putative substrates.

The Mdn1 MIDAS domain bears structural homology to the integrin MIDAS domain (*Chen et al., 2018*; *Garbarino and Gibbons, 2002*). Integrin has also been shown to form catch bonds, albeit with a fragment of its substrate (containing the RGD consensus sequence) rather than with two different full-length, folded proteins (*Astrof et al., 2006*; *Kong et al., 2009*; *Puklin-Faucher et al., 2006*; *Shimaoka et al., 2003*). Interestingly, the critical force at which bond lifetime is maximized is ~10–30 pN for integrin (*Chen et al., 2011*; *Kong et al., 2009*), a force-range relevant for meso-scale cell adhesions, in contrast to ~4 pN for Mdn1 MIDAS (*Figures 3G and 4F*), matching the forces that dynein, a AAA protein with high sequence similarity to Mdn1, is capable of producing (4.3 pN) (*Belyy et al., 2016*). We therefore suggest that MIDAS domains can be tuned to form catch bonds at forces ranging from the single motor to the multi-motor regime, as needed for specific cellular contexts. Toward this end, numerous AAA motor proteins including CbbQ (*Sutter et al., 2015*; *Tsai et al., 2015*; *Tsai et al., 2020*), Vwa8 (*Luo et al., 2017*), magnesium chelatase (*Fodje et al., 2001*), and others (*Iyer et al., 2004*), as well as other proteins with known mechanical activation such as von Willebrand factor (*Fredrickson et al., 1998*; *Kim et al., 2010*; *Ruggeri and Ware, 1993*), either contain or couple their activities with MIDAS domains. Further studies are needed to investigate the mechanoregulation of these MIDAS domains binding to their substrates.

The force jump assay developed for this study builds off of similar, previously reported force spectroscopy methods (*Cleary et al., 2014*; *Kilchherr et al., 2016*; *Kim et al., 2010*; *Kostrz et al., 2019*), with the benefit of being applicable to a wide range of bimolecular interactions (so long as the molecules can be conjugated to a DNA oligonucleotide or fused to GFP). The use of the bridge strand of DNA in our force jump assay enables the same single molecules to be measured more than once, and has the additional advantage of acting as a 'force divider', enabling smaller forces to be applied to the proteins of interest while holding a larger force of the dsDNA handles (thus holding them taut and maximizing spatial resolution). An associated drawback, however, is that applied forces on the proteins must be indirectly quantified using a mechanical circuit model (*Figure 3—figure supplement 4*). Similar to other experimental geometries used to measure catch bonds (*Guo and Guilford, 2006*; *Huang et al., 2017*), our force jump assay applies force at a very high rate (~20,000 pN/s), which may influence protein-protein binding (*Rao et al., 2019*). Nonetheless, we note that we use the same assay for all proteins in this study, and that the shortest binding durations we measure (20 ms) are >100-fold longer than the estimated rise time of the force jump. In future studies, the molecular configuration used here for the force jump assay (*Figure 2A*) can be reused for slower force ramp experiments to assess the force loading rate dependence of protein-protein binding durations.

Catch bonds are an example of how mechanical forces can directly regulate essential protein-ligand interactions. As opposed to other forms of regulation, such as phosphorylation, mechanoregulation is nearly instantaneous, and reversible without the need for accessory enzymes. Examples of known catch bonds range from cell adhesion through FimH (*Thomas et al., 2002*), selectins (*Marshall et al., 2003*), and vinculin (*Huang et al., 2017*), to outside-in signaling through integrins (*Kong et al., 2009*) and notch-jagged (*Luca et al., 2017*), to motor proteins including dynein (*Kunwar et al., 2011*) and myosin (*Guo and Guilford, 2006*), and even to kinetochore attachments during mitosis (*Akiyoshi et al., 2010*). Our data uncovers catch bonds between proteins involved in ribosome biogenesis, and, in contrast to these known examples, between a protein and more than one substrate.

# Materials and methods
## Protein expression and purification
The SNAP-MIDAS construct, as reported previously (*Mickolajczyk et al., 2020*), was generated by subcloning Mdn1 aa4381-4717 into pSNAP-tag(T7)–2 (NEB N9181S) downstream of the SNAP tag. An N-terminal 6xHis tag and a tobacco etch virus (TEV) protease site were added upstream of the SNAP tag. The Y4666R construct was generated using basic restriction cloning, verified by Sanger sequencing. Both MIDAS constructs were expressed and purified using the same protocol. MIDAS

proteins were expressed in *E. coli* Rosetta (DE3) pLysS cells (Merck 70954) grown in Miller's LB medium (Formedium LMM105). Expression was induced at A600 = 0.6–0.8 with 0.5 mM IPTG (Goldbio), and the cultures were held at 18°C for 16 hr. Cultures were pelleted and resuspended in lysis buffer (50 mM HEPES [pH 7.5], 150 mM NaCl, 1 mM $MgCl_2$, 10% w/v glycerol, 20 mM imidazole, 5 mM 2-mercaptoethanol, 1 mM PMSF, 3 U/mL benzonase, 1× Roche complete protease inhibitor without EDTA). All purification steps were carried out at 4°C. Cells were lysed using an Emulsiflex-C5 homogenizer (Avestin, 4 cycles at ~10,000 psi), and the crude lysate was centrifuged at 55,000 rpm in a Type 70 Ti rotor (Beckman Coulter) for 30 min. The supernatant was filtered through a 0.22 μm Millex-GP PES membrane (Millipore SLGP033RS) and loaded onto a HisTrap HP column (Cytiva 17524701) pre-equilibrated with wash buffer (50 mM HEPES [pH 7.5], 150 mM NaCl, 1 mM $MgCl_2$, 10% w/v glycerol, 20 mM imidazole, 5 mM 2-mercaptoethanol). The column was washed with 25 mL of wash buffer and eluted with wash buffer plus 400 mM imidazole. The eluent was treated with TEV protease (~0.1 mg/mL) and dialyzed against 1 L of dialysis buffer (20 mM HEPES [pH 7.5], 100 mM NaCl, 1 mM $MgCl_2$, 10% w/v glycerol, 5 mM 2-mercaptoethanol) for 3 hr at 4°C. The supernatant was then loaded onto a HiTrap Q HP column (Cytiva 29051325) and eluted over a gradient of low salt (same as dialysis buffer) and high salt (dialysis buffer with 1 M NaCl) buffers. Relevant factions were located by SDS-PAGE, pooled, and concentrated using a 30 kDa cutoff Amicon Ultra-4 centrifugal filter (Millipore UFC803008). The samples were then loaded onto a Superdex 200 Increase 10/300 GL column (Cytiva 28990944) in binding assay buffer (50 mM Tris [pH 7.5], 80 mM NaCl, 5 mM $MgCl_2$, 5% w/v glycerol, 0.002% Tween-20, 1 mM DTT). Concentration was determined using the colorimetric Bradford assay (Bio-Rad 5000006).

The plasmid containing the SNAP-tagged GFP nanobody was purchased on Addgene (#82711). This plasmid was transformed into Rosetta cells and prepared as described above. The same protocols and buffers were used up through elution from the HisTrap column. The His-eluent was concentrated using a 30 kDa cutoff Amicon Ultra-4 centrifugal filter (Millipore UFC803008), filtered using a 0.22 μm Millex-GP PES membrane (Millipore SLGP033RS), and loaded onto the Superdex 200 Increase 10/300 GL column (Cytiva 28990944) in storage buffer (20 mM HEPES [pH 7.5], 150 mM NaCl, 1 mM $MgCl_2$, 3% w/v glycerol, 1 mM DTT).

Rsa4 and Ytm1 were amplified from a *S. pombe* mitotic cDNA library (Cosmo Bio 02–703) and cloned into pAceBac1 (Geneva Biotech) with C-terminal GFP, TEV, and 6xHis tags Recombinant baculoviruses were generated using the Bac-to-Bac system (Thermo Fisher). High Five cells (Thermo Fisher B85502) were grown to ~3.0 million cells/mL in Express Five SFM (Thermo Fisher 10486025) supplemented with antibiotic-antimycotic (Life Technologies 15240–062) and 16 mM L-glutamine (Life Technologies 25030–081) prior to infection with P3 viral stocks at a 1:50 virus:media ratio. Cells were cultured in suspension (27°C, shaking at 115 rpm) for 48 hr prior to harvesting.

Rsa4-GFP purification was carried out at 4°C. Cells were lysed using a Dounce homogenizer (Thomas Scientific) in ~25 mL of lysis buffer (50 mM HEPES [pH 6.5], 400 mM NaCl, 1 mM $MgCl_2$, 10 % w/v glycerol, 20 mM imidazole, 5 mM 2-mercaptoethanol, 0.2 mM ATP, 1 mM PMSF, 3 U/mL benzonase, 1× Roche complete protease inhibitor without EDTA) per L of initial cell culture. The crude lysate was centrifuged at 55,000 rpm in a Type 70 Ti rotor (Beckman Coulter) for 1 hr then filtered using 0.22 μm Millex-GP PES membrane filters (Millipore SLGP033RS). The clarified lysate was loaded onto a HisTrap FF Crude column (Cytiva 29048631) pre-equilibrated with wash buffer (50 mM HEPES [pH 6.5], 400 mM NaCl, 1 mM $MgCl_2$, 10% w/v glycerol, 20 mM imidazole, 5 mM 2-mercaptoethanol). The column was washed with 25 mL wash buffer before elution with elution buffer (wash buffer with 400 mM imidazole). The eluent was treated with TEV protease (~0.1 mg/mL) and dialyzed against 1 L of dialysis buffer (20 mM HEPES [pH 6.5], 100 mM NaCl, 1 mM $MgCl_2$, 10% w/v glycerol, 5 mM 2-mercaptoethanol) for 3 hr at 4°C. The sample was then loaded onto a HiTrap SP HP column (Cytiva 29051324) pre-equilibrated with dialysis buffer, and eluted over a gradient of dialysis and elution (20 mM HEPES [pH 6.5], 1 M NaCl, 1 mM $MgCl_2$, 10% w/v glycerol, 5 mM 2-mercaptoethanol) buffers. Relevant fractions were pooled, concentrated using a 50 kDa Amicon Ultra-4 centrifugal filter (Millipore UFC805008), filtered using a 0.22 μm Millex-GP PES membrane (Millipore SLGP033RS), and loaded onto the Superdex 200 Increase 10/300 GL column (Cytiva 28990944) in binding assay buffer (50 mM Tris [pH 7.5], 80 mM NaCl, 5 mM $MgCl_2$, 5% w/v glycerol, 0.002% Tween-20, 1 mM DTT). Concentration was determined using the colorimetric Bradford assay (Bio-Rad 5000006). Rsa4-SNAP was generated and purified using the same methods as Rsa4-GFP.

Ytm1-GFP purification was carried out at 4°C. Cells were lysed using a Dounce homogenizer (Thomas Scientific) in ~25 mL of lysis buffer (50 mM HEPES [pH 7.5], 400 mM NaCl, 1 mM MgCl$_2$, 10 % w/v glycerol, 20 mM imidazole, 5 mM 2-mercaptoethanol, 0.2 mM ATP, 1 mM PMSF, 3 U/mL benzonase, 1× Roche complete protease inhibitor without EDTA) per L of initial cell culture. The crude lysate was centrifuged at 55,000 rpm in a Type 70 Ti rotor (Beckman Coulter) for 1 hr then filtered using 0.22 μm Millex-GP PES membrane filters (Millipore SLGP033RS). The clarified lysate was loaded onto a HisTrap FF Crude column (Cytiva 29048631) pre-equilibrated with wash buffer (50 mM HEPES [pH 7.5], 400 mM NaCl, 1 mM MgCl$_2$, 10% w/v glycerol, 20 mM imidazole, 5 mM 2-mercaptoethanol). The column was washed with 25 mL wash buffer before elution with elution buffer (wash buffer with 400 mM imidazole). The eluent was treated with TEV protease (~0.1 mg/mL) and dialyzed against 1 L of dialysis buffer (20 mM HEPES [pH 7.5], 100 mM NaCl, 1 mM MgCl$_2$, 10% w/v glycerol, 5 mM 2-mercaptoethanol) for 3 hr at 4°C. The sample was then loaded onto a HiTrap Q HP column (Cytiva 29051325) pre-equilibrated with dialysis buffer, and eluted over a gradient of dialysis and elution (20 mM HEPES [pH 7.5], 1 M NaCl, 1 mM MgCl$_2$, 10% w/v glycerol, 5 mM 2-mercaptoethanol) buffers. Relevant fractions were pooled, concentrated using a 50 kDa Amicon Ultra-4 centrifugal filter (Millipore UFC805008), filtered using a 0.22 μm Millex-GP PES membrane (Millipore SLGP033RS), and loaded onto the Superdex 200 Increase 10/300 GL column (Cytiva 28990944) in binding assay buffer (50 mM Tris [pH 7.5], 80 mM NaCl, 5 mM MgCl$_2$, 5% w/v glycerol, 0.002% Tween-20, 1 mM DTT). Concentration was determined using the colorimetric Bradford assay (Bio-Rad 5000006).

## Mass photometry

All mass photometry data were taken using a Refeyn OneMP mass photometer (Refeyn Ltd). Movies were acquired for 6000 frames (60 s) using AcquireMP software (version 2.4.0) and analyzed using DiscoverMP software (version 2.4.0, Refeyn Ltd), all with default settings. Proteins were measured by adding 2 μL of stock solution (100 nM) to an 8 μL droplet of filtered phosphate buffered saline (Gibco 14190144). Contrast measurements were converted to molecular weights using a standard curve generated with bovine serum albumin (Thermo 23210) and Urease (Sigma U7752). The data were fitted to a Gaussian in MATLAB (Mathworks, Natick, MA) to determine the measured molecular weight.

## Native mass spectrometry

The purified protein samples were buffer-exchanged into nMS solution (150 mM ammonium acetate, 0.01% Tween-20, pH 7.5) using Zeba desalting microspin columns with a 40 kDa molecular weight cutoff (Thermo Scientific). The buffer-exchanged sample was diluted to 2 μM and was loaded into a gold-coated quartz capillary tip that was prepared in-house. The sample was then electrosprayed into an Exactive Plus EMR instrument (Thermo Fisher Scientific) using a modified static nanospray source (*Olinares and Chait, 2020*). The MS parameters used included: spray voltage, 1.22 kV; capillary temperature, 200°C; S-lens RF level, 200; resolving power, 17,500 at m/z of 200; AGC target, 1 × 10$^6$; number of microscans, 5; maximum injection time, 200 ms; in-source dissociation, 100 V; injection flatapole, 8 V; interflatapole, 7 V; bent flatapole, 5–6 V; high energy collision dissociation, 10 V; ultra-high vacuum pressure, 6 × 10$^{-10}$ mbar; total number of scans, 100. Mass calibration in positive EMR mode was performed using cesium iodide. Raw nMS spectra were visualized using Thermo Xcalibur Qual Browser (version 4.2.47). Data processing and spectra deconvolution were performed using UniDec version 4.2 (*Marty et al., 2015*; *Reid et al., 2019*). The UniDec parameters used were m/z range: 2000–7000; mass range: 10,000–200,000 Da; sample mass every 1 Da; smooth charge state distribution, on; peak shape function, Gaussian; and Beta softmax function setting, 20.

## Size exclusion chromatography coelution assays

All coelution experiments were run in binding assay buffer (50 mM Tris [pH 7.5], 80 mM NaCl, 5 mM MgCl$_2$, 5% w/v glycerol, 0.002% Tween-20, 1 mM DTT) using a Superdex 200 Increase 10/300 GL column (Cytiva 28990944). Conditions run were MIDAS-WT alone, MIDAS-Y4666R alone, Rsa4-GFP alone, Rsa4-GFP with MIDAS-WT, Rsa4-GFP with MIDAS-Y4666R, Ytm1-GFP alone, Ytm1-GFP with MIDAS-WT, and Ytm1-GFP with MIDAS-Y4666R. In all cases, the MIDAS protein was at 60 μM and the Rsa4-GFP or Ytm1-GFP was at 40 μM in 500 μL total. Proteins were mixed and dialyzed against binding assay buffer for 3 hr prior to injection onto the column. The eluent was collected in 250 μL fractions,

which were analyzed both by SDS-PAGE and by GFP fluorescence. SDS-PAGE was performed using precast Novex 4–20% Tris-Glycine gels (Thermo Fisher XP04205BOX). Coomassie-stained gels were imaged using a LiCOR Odyssey system. Fluorescence measurements were made using a Synergy Neo Microplate reader (488 nm excitation, 528 nm emission).

## Microscale thermophoresis

Binding affinities between Mdn1 MIDAS-WT or MIDAS-Y4666R and Rsa4-GFP or Ytm1-GFP were measured using a Monolith NT.115 instrument (NanoTemper Technologies, Munich, Germany). All experiments were run in binding assay buffer (50 mM Tris [pH 7.5], 80 mM NaCl, 5 mM $MgCl_2$, 5% w/v glycerol, 0.002% Tween-20, 1 mM DTT), matching the buffer that all proteins were stored in. Rsa4-GFP or Ytm1-GFP were diluted to 100 nM and filtered using a 0.22 µm centrifugal filter (Millipore UFC30GV00), and mixed 1:1 with MIDAS protein (initial stock 80 µM, prepared in 1:2 serial dilutions) leading to a final concentration of 50 nM. Proteins were mixed for 5–10 min at room temperature in the dark before being loaded into Monolith NT.115 series premium capillaries (Nano-Temper MO-K025). Measurements were performed using 40–60% excitation power and the medium MST intensity option within the default settings in MO.Control software (NanoTemper Technologies, Munich, Germany). The raw fluorescence data were exported to MO.Affinity software and converted to $|\Delta F_{Norm}|$ by comparing the fluorescence before heating to the fluorescence 1.5 s after heating. $|\Delta F_{Norm}|$ as a function of MIDAS concentration could only be fitted successfully for MIDAS-WT with Rsa4-GFP, and were fitted with a binding isotherm in MATLAB (Mathworks, Natick, MA):

$$Y = \frac{Y_{Max} \cdot [S]}{K_D + [S]}$$

where $Y_{max}$ is the maximum system response. All MST measurements were repeated four times on four different days using at least two separate preparations of each of the proteins used.

## DNA substrates for optical tweezers force jump assay

The force jump assay tether consists of two 1.5 kilobase-pair biotinylated dsDNA handles connected by a ssDNA bridge. SNAP-GFP nanobody and SNAP-MIDAS were covalently attached to oligonucleotides that anneal to the bridge. The dsDNA handles are the same as were used in our previous study (*Mickolajczyk et al., 2021*). They were generated by PCR using primers designed to produce overhangs as needed to anneal the bridge stands. The forward primer for the 5' overhang handle contains an inverted base (iInvd) that terminates Phusion DNA polymerase (with a sequence of 5'-CAACCATGAGCACGAATCCTAAACCT/iInvdT/GCATAACCCCTTGGGGCCTCTAAACG-3'). The forward primer for the 3' overhang handle contains inverted bases at its 5' end that terminate Taq polymerase (with a sequence of 5'-/5InvdG/iInvdC//iInvdA//iInvdA//iInvdA//iInvdT//iInvdC//iInvdT//iInvdC//iInvdC//iInvdG//iInvdG//iInvdG//iInvdG//iInvdT//iInvdT//iInvdC//iInvdC//iInvdC//iInvdC//iInvdA//iInvdA//iInvdT//iInvdA//iInvdC//iInvdG/TAGTCTAGAGAATTCATTGCGTTCTGTACA/3ddC/–3'). The reverse primers for the 5' and 3' overhang handles contain a biotin at their respective 5' ends for bead attachment. The ssDNA bridges, which anneal to the overhangs, were synthesized by Integrated DNA Technologies (Coralville, IA). The 55 nt bridge has the sequence 5'-GGGAGACAACCATGAGCACGAATCCTAAACCTCCTCACTGTCTCGTCCGTCGTTCCGTCCTGT<u>CCTTTCCCCTCTCTTTTTCTTTCCCCTTTCTCCTCTCTCCTCTCCCTTCTCTCCCC</u>CTCACCTCACGTCCGCCAGATCCACAGTTCGTGTACAGAACGCAATGAATTCTCTAGACTA-3', and the 70 nt bridge has the sequence 5'-GGGAGACAACCATGAGCACGAATCCTAAACCTCCTCACTGTCTCGTCCGTCGTTCCGTCCTGT<u>CCTTTCCCCTCTCTTTTTCTTTCCCCTTTCTCCTCTCTCCTCTCCCTTCTCTCCCCCCTTTCTCCTCTCTCTCACCTC</u>ACGTCCGCCAGATCCACAGTTCGTGTACAGAACGCAATGAATTCTCTAGACTA-3'. In both cases the underlined region is the portion that remains single-stranded in the force jump assay.

The oligonucleotides used to connect SNAP-tagged proteins to the bridge were also synthesized by Integrated DNA Technologies (Coralville, IA) and have the sequences: 5'-Am-<u>CTCCTCTCTTTT</u>ACAGGACGGAAGGACAGACGAGAAAGTGAG-3' and 5'-GAACTGTGGATCTGGCGGACGTGAGGTGAG<u>TTTCTCCTTTCT</u>-Am-3', where the underline regions are the portions that do not anneal to the bridge, and Am denotes modification with a free amino group. A benzylguanine (BG) group was covalently attached to the amino-oligonucleotides using BG-GLA-NHS (New England BioLabs S9151S) under manufacturer-suggested reaction conditions, and subsequently purified using Micro Bio-Spin

P-6 gel columns (Bio-Rad 7326221). The BG-oligos were then conjugated to SNAP-proteins by mixing at a 2:1 molar ratio in storage buffer (20 mM HEPES [pH 7.5], 150 mM NaCl, 1 mM MgCl$_2$, 3% w/v glycerol, 1 mM DTT) incubating in the dark at 4°C overnight. Excess oligonucleotides and unreacted SNAP proteins were removed by purification on a Superdex 200 Increase 10/300 GL column (Cytiva 28990944) in storage buffer. Fractions with oligo-protein conjugates were identified via SDS-PAGE, pooled, and flash-frozen in liquid nitrogen prior to use.

To prepare for experiments, the 5' overhang dsDNA handle and bridge strands were first annealed by mixing 50 nM of 5' overhang handles with 12.5 nM of bridge ssDNA in storage buffer, incubating at 65°C for 1 min, then cooling to 4°C at 0.1°C/s. This annealed product was then mixed with oligo-SNAP-nanobody and oligo-SNAP-MIDAS (final 2 nM, 10 nM, 10 nM, respectively) in optical tweezers assay buffer (50 mM Tris [pH 7.5], 80 mM NaCl, 5 mM MgCl$_2$, 0.1 mg/mL BSA, 0.002% Tween-20, 1 mM DTT) and incubated on ice for 30 min. This reaction was then diluted 5-fold into optical tweezers assay buffer containing 0.08% w/v streptavidin-coated polystyrene beads (2.1 µm diameter, Sphero-tech) and incubated on ice for at least 15 min. This mixture was diluted 1000-fold in optical tweezers assay buffer at the time of use. Separately, 5 nM 3' overhand handles were mixed with 0.08% w/v streptavidin-coated polystyrene beads (2.1 µm diameter, Spherotech) in optical tweezers assay buffer, and incubated on ice for at least 15 min. This mixture was also diluted 1000-fold in optical tweezers assay buffer at the time of use.

To form a tether, a bead conjugated to the 5' overhang handle complex was immobilized on a pipette via suction and brought into close proximity to a second bead conjugated to 3' overhang handles held in an optical trap. Upon hybridization of the 3' overhang handle with the 5' overhang handle complex, a 'tether' was formed. For both the 55 and 70 nt bridge constructs, we ensured that only a single tether was drawn between each pair of beads by measuring the force-extension curve (in the range of 4–20 pN) and matching it to a well-characterized hairpin construct held between identical dsDNA handles (*Figure 2D*; *Mickolajczyk et al., 2021*). This hairpin standard was chosen because it can be mechanically unwound at ~20 pN, thus making it easy to identify if a single tether is drawn (if more than one tether is drawn, >20 pN is needed to observe hairpin unwinding). The force-extension curves are dominated by the dsDNA handles in the 4–20 pN regime. Thus, for the 55 and 70 nt bridge constructs, instances with multiple tethers (i.e. multiple connected sets of dsDNA handles) produced force-extension curves that significantly deviated from (i.e. could not be superimposed onto) that of the hairpin standard.

## Optical tweezers force jump assay

The force jump assay developed for this study is related to the ReaLiSM and junctured-DNA-tweezers approaches (*Kim et al., 2010*; *Kostrz et al., 2019*), as well as other conceptually similar approaches (*Cleary et al., 2014*; *Kilchherr et al., 2016*). All single-molecule measurements were made using an optical tweezers instrument ('MiniTweezers') (*Smith et al., 2003*). Optical tweezers assay buffer (50 mM Tris [pH 7.5], 80 mM NaCl, 5 mM MgCl$_2$, 0.1 mg/mL BSA, 0.002% Tween-20, 1 mM DTT) was used for all experiments. At the start of each experiment, a tether was formed in the sample chamber and its force-extension behavior was obtained by moving the optically trapped bead at a speed of 70 nm/s away from the second bead held in the pipette. Rsa4-GFP or Ytm1-GFP were injected using a shunt line to a final of 20 nM before each tether was pulled. The experiment was run by using constant force mode and switching between 0.5 pN low force and 4–12 pN high force. All force levels were held for at least 5 s (empirically determined to be long enough to include all events). Force feedback was temporarily turned off at the low force to prevent large position fluctuations. Data was collected at 200 Hz (instrument response time ~5–10 ms). Data was collected on individual molecules for 15 min or less. In some cases, data collection on a given molecule was prematurely terminated due to instrument drift out of focus or inadvertent mechanical separation of the bridge DNA from the handles while being held at the high-force level. All experiments were run at 23°C ± 1 °C.

Single-molecule data were analyzed in MATLAB (Natick, MA). High-force positional plateaus were investigated using a two-state hidden Markov model, which was modified from existing code (*Mickolajczyk et al., 2015*). The data were centered at 0 by subtracting the mean of the first five data points from the whole vector, and the effective standard deviation was determined by building a distribution of 10-data point boxcar standard deviations (eSD), and finding the median. The intermediate (proteins bound) and final (proteins unbound) positions on the high-force plateau were treated as Gaussian

emitters, with centers 0 and +4 eSD units, and standard deviations one eSD unit. A transition matrix was constructed to only allow transitions from the zero-position state to the higher position state. The emission matrix, transition matrix, and experimental data were used as input to the Viterbi algorithm (*Viterbi, 1967*), which returned the most likely sequence of hidden states. If a state change was detected, then the bond lifetime was calculated as the duration preceding the state change, and Δx was calculated as the difference in means between an equal number of data points (corresponding to the binding duration) before and after the state change. In some cases, there was drift in the data, which could lead to a larger position change than the Δx jump over time. In these cases, the high-force plateau was truncated to the area just around the Δx jump. Events were only accepted if they were at least four data points long. Every event was manually inspected, and only events that were clearly an instantaneous change in motion (rather than drift) were retained for further analysis. Pooled values of bond lifetime data were fitted to a single-exponential:

$$Survival\ Probability = Ae^{bx}$$

## Mechanical circuit model

All spring elements in the mechanical circuit model were modeled using the worm-like chain model:

$$\frac{FL_p}{k_BT} = \frac{1}{4}\left(1 - \frac{x}{L_c}\right)^{-2} - \frac{1}{4} + \frac{x}{L_c}$$

where $L_c$ is the contour length and $L_p$ is the persistence length. The parameters used were persistence length 1 nm, and contour length 0.59 nm per DNA nucleotide (*Liphardt et al., 2001*). The bridge strand was 55 or 70 nt in length, and the loading strand was 24 nt in length. When both springs were engaged (proteins bound), the force-extension curve of the system was:

$$F_{Tot}\left(x\right) = F_{Bridge}\left(x + L_{prot}\right) + F_{Load}\left(x\right)$$

When the proteins dissociated, the force-extension curve of the system was:

$$F_{Tot}\left(x\right) = F_{Bridge}\left(x + L_{prot}\right)$$

where $L_{prot}$ is the inextensible length of the proteins. Based on existing structural data, we estimate $L_{prot}$ to be 8 nm (*Ahmed et al., 2019*). For a given total force placed on the system ($F_{Tot}$), the force placed on the proteins ($F_{Load}$) and the theoretical Δx could be calculated. All data analysis and fitting was performed in MATLAB (Natick, MA).

# Acknowledgements

We thank members of the Kapoor and Liu Labs for useful discussions. We thank Jun Funabiki and the Paul Nurse Lab for generously providing the *S. pombe* mitotic cDNA and Xiaocong Cao (Liu Lab) for providing DNA handle reagents and protocols. We thank the Rockefeller High-Throughput Screening Resource Center for assistance with the MST experiments. TMK is grateful to the NIH/NIGMS for funding (R35 GM130234-01). BTC is grateful to NIH/NIGMS for funding (P41 GM109824 and P41 GM103314). SL is supported by the Robertson Foundation and NIH (DP2HG010510). KJM is supported by a National Cancer Institute K00 Fellowship (K00CA223018).

# Additional information

## Funding

| Funder | Grant reference number | Author |
|---|---|---|
| National Institutes of Health | R35 GM130234-01 | Tarun Kapoor |
| National Institutes of Health | P41 GM109824 | Brian T Chait |

| Funder | Grant reference number | Author |
|---|---|---|
| National Institutes of Health | P41 GM103314 | Brian T Chait |
| National Institutes of Health | DP2HG010510 | Shixin Liu |
| Robertson Foundation | | Shixin Liu |
| National Cancer Institute | K00CA223018 | Keith J Mickolajczyk |

The funders had no role in study design, data collection and interpretation, or the decision to submit the work for publication.

## Author contributions

Keith J Mickolajczyk, Conceptualization, Data curation, Formal analysis, Funding acquisition, Investigation, Methodology, Resources, Software, Validation, Visualization, Writing – original draft, Writing – review and editing; Paul Dominic B Olinares, Data curation, Formal analysis, Writing – review and editing; Brian T Chait, Shixin Liu, Funding acquisition, Project administration, Resources, Writing – review and editing; Tarun M Kapoor, Conceptualization, Funding acquisition, Investigation, Project administration, Resources, Supervision, Writing – original draft, Writing – review and editing

## Author ORCIDs

Keith J Mickolajczyk  http://orcid.org/0000-0001-9445-0325
Shixin Liu  http://orcid.org/0000-0003-4238-7066
Tarun M Kapoor  http://orcid.org/0000-0003-0628-211X

## Decision letter and Author response

Decision letter https://doi.org/10.7554/eLife.73534.sa1
Author response https://doi.org/10.7554/eLife.73534.sa2

# Additional files

## Supplementary files

• Transparent reporting form

## Data availability

All data generated or analyzed during this study are included in the manuscript and source files. Four Excel files including source data for figures 1, 2, 3, 4 and supplements. Raw TIFF files for all gels in figures 1, 3, 4 provided in zipped folders. One MATLAB script that runs the mechanical circuit model used in figures 3 and 4.

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
