## [Editor Report]

Mickolajczyk et al., use solution and single-molecule approaches to characterize the binding of the MIDAS domain from the ribosome maturation factor Mdn1 with ubiquitin-like domains from two assembly factors, Ytm1 and Rsa4. Both interactions are specific but weak in solution. A clever experimental setup allows the authors to also measure the interaction with optical tweezers, revealing catch and slip bond modes, depending on the applied load. The off-rate is lowest at ~4 pN. The behavior might help to explain how Mdn1 binding to Ytm1 and Rsa4 (and possibly other UBL-containing proteins) is stable enough for protein extraction from pre-60S particles without excessive idle binding of free assembly factors.

---

## [Decision Letter]

**Decision letter after peer review:**

Thank you for submitting your article “The MIDAS domain of AAA mechanoenzyme Mdn1 forms catch bonds with two different substrates” for consideration by *eLife*. Your article has been reviewed by 3 peer reviewers, and the evaluation has been overseen by a Reviewing Editor and Cynthia Wolberger as the Senior Editor. The following individual involved in review of your submission has agreed to reveal their identity: Helgo Schmidt (Reviewer #2).

Essential revisions:

From reviewer 1:

For force jump experiments, the GFP-nanobody bound to one DNA handle holds Ras4-GFP or Ytm1-GFP in place. Within error, the force-dependent bond lifetimes do look very similar for Rsa4 and Ytm1 (Figure 4F). Can the authors rule out that this non-covalent interaction (between the GFP and the GFP-nanobody) is in fact the one that has catch-bond character? This possibility is perhaps not very likely, but since it directly affects the central conclusion of the manuscript, it would be good if a control experiment can be done to rule out this possibility.

*Reviewer #1 (Recommendations for the authors):*

I have a few points that I think should be addressed prior to publication, detailed below.

(1) For force jump experiments, the GFP-nanobody bound to one DNA handle holds Ras4-GFP or Ytm1-GFP in place. Within error, the force-dependent bond lifetimes do look very similar for Rsa4 and Ytm1 (Figure 4F). Can the authors rule out that this non-covalent interaction (between the GFP and the GFP-nanobody) is in fact the one that has catch-bond character? This possibility is perhaps not very likely, but since it directly affects the central conclusion of the manuscript, I am wondering if a control experiment can be done to rule out this possibility.

(2) The model for how force regulates MIDAS binding is very elegant, but I do not fully understand the description in the discussion. Is the idea that the unstructured linker contributes to the tension that then establishes a long-lived interactions between the MIDAS and the UBL domains? If so, wouldn’t the (entropic) cost of stretching the unstructured linker disfavor binding of the MIDAS to the UBL domain? Also, the bond lifetimes are quite short, about 1 second even at the maximum (at ~4 pN). What are the kinetics of Rsa4/Ytm1 removal? Is the observed increase in bond lifetime expected to significantly enhance the process?

(3) While the catch bond character of the interaction studied here is compellingly demonstrated, the fits of a single-exponential model to the measured survival probabilities (Figure 3 and S3) does not look great. There seems to be some systematic deviation, but that is difficult to evaluate because the authors do not show the residuals. The authors should show the residuals for all fits and demonstrate that the quality of the fit does not improve when a different model is used. If in fact a different model provides a significantly better fit, the interpretation of how the molecular interaction is disrupted might change.

(4) I do not understand Figure 1D. How the force-extension curve shown in Figure 2D “ensured that only a single “tether” was drawn between each pair of beads” (Methods)? What is meant by Δ(Extension) on the x axis? The extension for the dsDNA handles used here (3 kb) should be much longer that the ~100 nm in Figure 1D.

(5) The 95% confidence intervals for the rate parameters (k0_c and k0_s) are larger than the mean. I assume that the confidence intervals would be substantially more narrow when more events are analyzed, which may be helpful when trying to understand the function of the Mdn1 quantitatively/kinetically. The assay design should make it possible to detect many events per molecule. Is there any reason why the number of events are moderately low (typically not exceeding 30)?

(6) In addition to reporting the number of events, the authors should also report the number of individual molecules that were analyzed for each condition.

(7) The assay design for the force jump experiments is very elegant. The use of a bridge or leash to study interactions is not completely novel. Conceptually similar approaches have been reported before, for instance by Kilchherr et al., (PMID: 27609897) and Kostrz et al., (PMID: 31548690). The authors should probably cite these studies.

*Reviewer #2 (Recommendations for the authors):*

– A clear drawback of this study is that the forces between the MIDAS domain and the UBL-Rsa4 and UBL-Ytm1 domains can only be indirectly quantified via "Mechanical circuit" modelling.

*Reviewer #3 (Recommendations for the authors):*

The manuscript by Mickolajczyk et al., reports on the development of a new optical tweezers-based unbinding-force assay to characterize the bond between the MIDAS domain of the mechanoenzyme Mdn1 and the ubiquitin-like (UBL) domain-containing ribosomal proteins Rsa1 and Ytm1. The authors show that while the affinity between the MIDAS domain and Rsa1 and Ytm1 is only weak ({greater than or equal to} 7 μM) in solution, it increases significantly under an applied load. Using the developed assay, the authors show that the bond between MIDAS and Rsa1/Ytm1 can be best explained by a catch-slip bond behavior. The authors suggest that the catch bonding between MIDAS and UBL domains plays a key role in the Mdn1-mediated ribosomal biogenesis. The reported results are highly interesting for the ribosomal and single-molecule biophysics communities and the developed DNA-tether-based optical tweezers assay will be useful for characterizing other molecular bonds. I believe the manuscript is appropriate for *eLife* and would recommend publication.

---

## [Author Response]

Essential revisions:From reviewer 1:For force jump experiments, the GFP-nanobody bound to one DNA handle holds Ras4-GFP or Ytm1-GFP in place. Within error, the force-dependent bond lifetimes do look very similar for Rsa4 and Ytm1 (Figure 4F). Can the authors rule out that this non-covalent interaction (between the GFP and the GFP-nanobody) is in fact the one that has catch-bond character? This possibility is perhaps not very likely, but since it directly affects the central conclusion of the manuscript, it would be good if a control experiment can be done to rule out this possibility.Reviewer #1 (Recommendations for the authors):I have a few points that I think should be addressed prior to publication, detailed below.(1) For force jump experiments, the GFP-nanobody bound to one DNA handle holds Ras4-GFP or Ytm1-GFP in place. Within error, the force-dependent bond lifetimes do look very similar for Rsa4 and Ytm1 (Figure 4F). Can the authors rule out that this non-covalent interaction (between the GFP and the GFP-nanobody) is in fact the one that has catch-bond character? This possibility is perhaps not very likely, but since it directly affects the central conclusion of the manuscript, I am wondering if a control experiment can be done to rule out this possibility.

We appreciate this point, and agree that we can strengthen our paper by ruling out the possibility that the GFP-GFP nanobody interaction contributes to the observed force-dependent bond lifetimes. To address this concern, we redesigned the experiment and established a control with a completely different ‘linking’ strategy. Briefly, we generated Rsa4 with a C-terminal SNAP tag (hereafter Rsa4-SNAP) instead of Rsa4-GFP. Rsa4SNAP was covalently attached to a DNA oligonucleotide (similarly to how SNAP-GFP nanobody was), and then annealed to the bridge DNA strand in the force jump assay (thus bypassing the need for GFP/ GFP nanobody). We measured the force-dependent bond lifetime of Rsa4-SNAP with SNAP-MIDAS across the same force range as Rsa4GFP with SNAP-MIDAS. Gratifyingly, we found that the fit for the Rsa4-GFP dataset could be overlaid on the Rsa4-SNAP dataset and hits each datapoint within error. We believe that this control experiment shows that GFP-GFP nanobody connection is not a significant contributing factor to the observed force-dependent bond lifetime measurements in this paper. These new data are including in supplements to Figure 3 (Figure 3 figure supplement 5 and 6).

The specific changes we made to the text are:

p. 7

“As an additional control, we repeated the force jump experiments using Rsa4 with a C-terminal SNAP tag (Rsa4-SNAP), which was directly conjugated to a ssDNA oligonucleotide. Datasets generated with Rsa4-GFP and Rsa4-SNAP were in good agreement with each other (Figure 3 figure supplements 5-6), indicating that the GFP-GFP nanobody connection did not influence bond lifetime measurements.”

(2) The model for how force regulates MIDAS binding is very elegant, but I do not fully understand the description in the discussion. Is the idea that the unstructured linker contributes to the tension that then establishes a long-lived interactions between the MIDAS and the UBL domains? If so, wouldn't the (entropic) cost of stretching the unstructured linker disfavor binding of the MIDAS to the UBL domain? Also, the bond lifetimes are quite short, about 1 second even at the maximum (at ~4 pN). What are the kinetics of Rsa4/Ytm1 removal? Is the observed increase in bond lifetime expected to significantly enhance the process?

We understand the reviewer’s concerns, and in response we have revised our Discussion improve the clarity and detail of our model. Our previously-published biochemical assays showed that MIDAS binding to the Mdn1 AAA ring occurred more readily as a bimolecular interaction (two domains prepared separately) than as an intramolecular interaction. We coupled this with Brownian Dynamics modeling to show that, indeed, there is an entropic cost to stretching out the unstructured linker in order to reach docking site (we approximate 1-2 pN of tension). This then set up a paradox – if intramolecular docking of the MIDAS domain is entropically disfavored, then how come the docked complex (on the pre-ribosome surface) is so stable? We speculate that the catch bond we characterize in the current work may, at least in part, resolve this paradox – tension along the unstructured linker would actually promote, rather than disfavor, MIDAS-UBL binding. We have also updated the diagram in Figure 4G in order to help explain our model in more detail.

With respect to the kinetics of removal, very little is currently known. However, we do know that when Mdn1 activity is inhibited using the small molecule Rbin-1, that ribosome biogenesis stalls altogether and pre-60S species build up in the nucleolus (Kawashima et al., *Cell*, 2015). Hence, the kinetics of ribosome assembly in the absence of Mdn1 must be very slow. We have revised the text to make it clearer that more work is needed in order to understand the kinetics and enzymology of Mdn1-driven steps in ribosome biogenesis.

Specific changes to the text: p. 9

“Based on our findings, we propose a model where the Mdn1 MIDAS-UBL catch bond plays a key role in assembly factor processing (Figure 4G). Mdn1 binds the pre-60S particle and establishes a tripartite interaction between its AAA ring, its MIDAS domain, and the UBL domain of the assembly factor substrate that it must extract (Chen et al., 2018; Kater et al., 2020; Sosnowski et al., 2018). In this MIDAS-docked state, Mdn1’s unstructured linker, which acts as an entropic spring, is stretched; computational modeling of MIDAS docking estimates that 1-2 pN of tension is built up and propagated along the MIDAS-UBL bond axis (Mickolajczyk et al., 2020). A catch bond mechanism helps to explain how this tension promotes, rather than disfavors, MIDAS-UBL binding. 1-2 pN is below the ~3.3-4.2 pN peak of the MIDAS-UBL catch bond, but one possibility is that additional tension is generated by Mdn1’s ATPase activity. The MIDAS domain remains bound to the UBL domain until the assembly factor dissociates from the pre-60S particle. Subsequently, the MIDAS domain undocks from the AAA ring, relieving tension across the bond with the UBL domain, deactivating the catch bond and facilitating assembly factor release. Ytm1 and Rsa4 removal occur at separate stages of pre-60S maturation, in the nucleolus and nucleus, respectively (Bassler et al., 2010; Kressler et al., 2012). We propose that our model holds for both these scenarios. We also note that free copies of Ytm1 and Rsa4 are present in the nucleolus and nucleus (Kressler et al., 2012); the Mdn1 catch bond thus also prevents Mdn1-Rsa4/Ytm1 binding while off the pre-60S, which would compete away usable copies of each protein. Currently, the kinetics of Mdn1-driven steps in ribosome assembly are not well-established. Future work is thus necessary to place the ~1 second bond lifetimes measured here within the multi-step enzymatic process of ribosome biogenesis.”

(3) While the catch bond character of the interaction studied here is compellingly demonstrated, the fits of a single-exponential model to the measured survival probabilities (Figure 3 and S3) does not look great. There seems to be some systematic deviation, but that is difficult to evaluate because the authors do not show the residuals. The authors should show the residuals for all fits and demonstrate that the quality of the fit does not improve when a different model is used. If in fact a different model provides a significantly better fit, the interpretation of how the molecular interaction is disrupted might change.Also Reviewer 31. Page 6: The authors write that the measured lifetime distributions could be fitted to a single exponential distribution, suggesting that the kinetics are best described by the unbinding from a single bound state. Judged from the data shown in Figure 3 A, B and C, there is a systematic deviation from a single exponential (a distinct shoulder is visible in the 1-2 second time range). The author should discuss this observation.

We agree with the reviewers, and in response we have (1) collected a larger data set, (2) included a new supplemental figure (Figure 3 figure supplement 1) comparing single versus double exponential fitting (adding a second exponential reduces the RMSE of the fit by <10%), and (3) Added plots of residuals (as well as QQ plots to show exponential fit quality) for every data set in the paper (see Figure 3 figure supplements 2, 3, and 6 as well as Figure 4 figure supplements 4 and 5). While plotting survival probability on a semilog Y axis has the advantage of enabling easy distinguishment between single exponentials (which look linear) and biexponentials (which look like two connected lines with different slopes), there is the drawback that the final 10% of measurements take up half the Y-axis (and thus noise looks more severe on the bottom 10% than it does on the top 90%). Increasing the number of measurements is the solution to this issue that we have favored. Overall, we believe our larger data set and our new series of figure supplements showing residuals help strengthen our observation that a single exponential is sufficient to fit the lifetime distributions.

Specific changes to the text (other than the new figures and their legends, as listed above):

p. 5

“The distribution (n=57 events from 14 molecules) of bond lifetimes could be fitted to a single exponential (appears linear on a semi-log plot) as opposed to a higher-order exponential (Figure 3—figure supplement 1), consistent with the kinetics of exit from a single bound state (Guo and Guilford 2006).”

p. 6

“All measured distributions could be fitted to a single exponential (residuals and analysis in Figure 3—figure supplements 2-3), indicating that applied force influenced the kinetics of dissociation, not the number of states from which dissociation could occur (Huang et al., 2017).”

(4) I do not understand Figure 1D. How the force-extension curve shown in Figure 2D "ensured that only a single "tether" was drawn between each pair of beads" (Methods)? What is meant by Δ(Extension) on the x axis? The extension for the dsDNA handles used here (3 kb) should be much longer that the ~100 nm in Figure 1D.

We understand the reviewer’s point, and in response we have revised the text and provide a more detailed explanation of Figure 1D. We define a tether as the entire construct (including both dsDNA handles and the bridge DNA) that connects the two beads. Experimentally, it is possible to inadvertently connect the two beads with multiple tethers rather than just one. Beads connected with more than one tether have a significantly different force-extension curve than beads connected by a single tether.

We use the hairpin construct as a standard because it is known to denature at ~20 pN applied force. For this construct, if multiple tethers connect the two beads, then >>20 pN of force is needed to observe hairpin denaturation. The portion of the force-extension curve in the <20 pN range is dominated by the dsDNA handles (the same dsDNA handles are used for all experiments). The 55 nt and 70 nt bridge constructs also have force-extension curves that are dominated by the dsDNA handles (since the handles are much longer than the bridge). Thus, if the force-extension curve of the 55 or 70 nt bridge constructs overlapped with that of the hairpin construct in the <20 pN range (specifically, we worked in the 4-20 pN range for simplicity), we concluded that a single tether connected to two beads. If we saw that the force-extension curve did not overlap with that of the hairpin construct in the 4-20 pN range, then we deemed it ineligible for measuring single-molecule bond lifetimes.

As the reviewer notes, the contour length of the DNA handles is much greater than the ~150 nm shown in Figure 2D. This is because we only measured in the range of 4-20 pN. We refer to the X-axis as ΔExtension rather than extension because it is not possible to experimentally find the true point of 0-nm extension (the two beads crash into each other). Instead, we measure the force-extension curve between 4-20 pN of force (ΔExtension=0 when F=4 pN). This annotation is consistent with our previous publications (Mickolajczyk et al., *Biophysical Journal*, 2021). We have revised the Figure 2D caption to define ΔExtension more explicitly.

Specific changes to the text:

p. 5

“For both bridge constructs, we ensured that only a single “tether” was drawn between each pair of beads by examining the shape of the force-extension curve and matching it to a known hairpin standard (Figure 2D, see Methods).”

Figure 2D caption

“(D) Example force-extension curves of the DNA handles connected by the 55 nt (red) and 70 nt (blue) bridge. Also shown is a hairpin (gray) which anneals to the dsDNA handles with a 20 nt region leftover; here the distinct unfolding/refolding pattern at loads above 20 pN enables identification of a single “tether” (two dsDNA handles connected by a bridge) between the two beads. Data generated on tethers whose force-extension curves did not overlap with that of the hairpin standard in the 4-20 pN (ΔExtension=0 when Force=4 pN) range were not used.”

Methods (p. 18)

“For both the 55 and 70 nt bridge constructs, we ensured that only a single tether was drawn between each pair of beads by measuring the force-extension curve (in the range of 4-20 pN) and matching it to a well-characterized hairpin construct held between identical dsDNA handles (Figure 2D) (Mickolajczyk et al., 2021). This hairpin standard was chosen because it can be mechanically unwound at ~20 pN, thus making it easy to identify if a single tether is drawn (if more than one tether is drawn, >20 pN is needed to observe hairpin unwinding). The force-extension curves are dominated by the dsDNA handles in the 4-20 pN regime. Thus, for the 55 and 70 nt bridge constructs, instances with multiple tethers (i.e. multiple connected sets of dsDNA handles) produced force-extension curves that significantly deviated from (i.e. could not be superimposed onto) that of the hairpin standard.”

(5) The 95% confidence intervals for the rate parameters (k0_c and k0_s) are larger than the mean. I assume that the confidence intervals would be substantially more narrow when more events are analyzed, which may be helpful when trying to understand the function of the Mdn1 quantitatively/kinetically. The assay design should make it possible to detect many events per molecule. Is there any reason why the number of events are moderately low (typically not exceeding 30)?

The reviewer is correct – the large confidence intervals are partially due to the number of counted events. In response, we have collected a larger data set for this revision, focusing specifically on Rsa4-GFP with the 55 nt bridge construct. The fit to the catch-slip model has improved substantially as a result:

kc0 = 5.3 ± 2.6 s^-1^, xc = -2.0 ± 0.8 nm, ks0 = 0.01 ± 0.02 s^-1^, and xs = 4.1 ± 1.6 nm (fit ± 95% CI).

As the reviewer pointed out, our assay design allows up to collect multiple events per molecule. As shown in Figure 2, we typically collect ~1 event per minute. Ultimately, the data collection bandwidth is limited to how long we can collect on a single set of molecules, which we found to be on the timescale of minutes (≤15 minutes spent per molecule). In some cases, data collection on a given molecule was prematurely terminated due to instrument drift out of focus or inadvertent mechanical separation of the bridge DNA from the handles while being held at the “high force” level. Thus, we can increase the number of events by collecting from a larger number of molecules, but cannot easily reach events counts in the hundreds for the twenty experimental conditions (five high force levels, two bridge constructs, two UBL proteins) as well as control conditions. We have revised the text to make limitations on data collection clearer.

Specific changes to the text:

Figure 3 caption

“A. Distributions of MIDAS-Rsa4 bond lifetimes (6 pN total force, 55 nt bridge construct) with either MIDAS-WT (n=57 events from 14 molecules) or MIDAS-Y4666R (n=29 events from 17 molecules). Survival probability is defined as one minus the empirical cumulative density function. For all distributions, the final data point was moved from y=0 to y=0.01 to enable semilog plotting. Black lines show fits to a single exponential.

B. Distributions of MIDAS-Rsa4 bond lifetimes on the 55 nt bridge construct at 4 pN (n=26 events from 8 molecules), 6 pN (n=57 events from 14 molecules), and 12 pN (n=29 events from 5 molecules) total applied force.”

p. 19

“Data was collected on individual molecules for fifteen minutes or less. In some cases, data collection on a given molecule was prematurely terminated due to instrument drift out of focus or inadvertent mechanical separation of the bridge DNA from the handles while being held at the high force level.”

(6) In addition to reporting the number of events, the authors should also report the number of individual molecules that were analyzed for each condition.

We agree with the reviewer, and have updated the text to include the number of molecules analyzed. These numbers are listed both in the captions for the main Figures 3 and 4, as well as for Figure 3 figure supplements 2, 3, and 5 and Figure 4 figure supplements 4 and 5. We note that in some cases, events at more than one force level were obtained from the same molecule. We have pasted the updated Figure 3 caption.

(7) The assay design for the force jump experiments is very elegant. The use of a bridge or leash to study interactions is not completely novel. Conceptually similar approaches have been reported before, for instance by Kilchherr et al., (PMID: 27609897) and Kostrz et al., (PMID: 31548690). The authors should probably cite these studies.

We thank the reviewer for mentioning these papers. We had cited the Kostrz et. al paper, but were unaware of the Kilchherr et al., paper (it was a nice read!). We have updated the citations both in the Methods section, and in a new Discussion paragraph (focused on comparing the techniques).

Specific changes to the text:

p. 10

“The force jump assay developed for this study builds off of similar, previously-reported force spectroscopy methods (Cleary et al., 2014; Kilchherr et al., 2016; Kim et al., 2010; Kostrz et al., 2019), with the benefit of being applicable to a wide range of bimolecular interactions (so long as the molecules can be conjugated to a DNA oligonucleotide or fused to GFP).”

p. 18

“The force jump assay developed for this study is related to the ReaLiSM and junctured-DNA-tweezers approaches (Kim et al., 2010; Kostrz et al., 2019), as well as other conceptually similar approaches (Kilchherr et al., 2016; Cleary et al., 2014).”

Reviewer #2 (Recommendations for the authors):– A clear drawback of this study is that the forces between the MIDAS domain and the UBL-Rsa4 and UBL-Ytm1 domains can only be indirectly quantified via "Mechanical circuit" modelling.

We appreciate this point, and in response we have written a new section of the Discussion comparing the strengths and drawbacks of the force jump assay. We explicitly mention the drawback of indirect force quantification.

Specific changes to the text:

p. 10-11

“The force jump assay developed for this study builds off of similar, previously-reported force spectroscopy methods (Cleary et al., 2014; Kilchherr et al., 2016; Kim et al., 2010; Kostrz et al., 2019), with the benefit of being applicable to a wide range of bimolecular interactions (so long as the molecules can be conjugated to a DNA oligonucleotide or fused to GFP). The use of the bridge strand of DNA in our force jump assay enables the same single molecules to be measured more than once, and has the additional advantage of acting as a “force divider”, enabling smaller forces to be applied to the proteins of interest while holding a larger force of the dsDNA handles (thus holding them taut and maximizing spatial resolution). An associated drawback, however, is that applied forces on the proteins must be indirectly quantified using a mechanical circuit model (Figure 3—figure supplement 4). Similar to other experimental geometries used to measure catch bonds (Guo and Guilford, 2006; Huang et al., 2017), our force jump assay applies force at a very high rate (~20,000 pN s^-1^), which may influence protein-protein binding (Rao et al., 2019). Nonetheless, we note that we use the same assay for all proteins in this study, and that the shortest binding durations we measure (20 ms) are >100-fold longer than the estimated rise time of the force jump. In future studies, the molecular configuration used here for the force jump assay (Figure 2A) can be reused for slower force ramp experiments to assess the force loading rate dependence of protein-protein binding durations.”